# Deep Learning-Based Technique for Remote Sensing Image Enhancement Using Multiscale Feature Fusion

**Ming Zhao** [ID]**, Rui Yang, Min Hu * and Botao Liu**

School of Computer Science, Yangtze University, Jingzhou 434023, China; hitmzhao@gmail.com (M.Z.);
201900912@yangtzeu.edu.cn (R.Y.); liubotao920@yangtzeu.edu.cn (B.L.)
**\*** Correspondence: huxongliang@gmail.com

**Abstract:** The present study proposes a novel deep-learning model for remote sensing image enhancement. It maintains image details while enhancing brightness in the feature extraction module. An improved hierarchical model named Global Spatial Attention Network (GSA-Net), based on U-Net for image enhancement, is proposed to improve the model's performance. To circumvent the issue of insufficient sample data, gamma correction is applied to create low-light images, which are then used as training examples. A loss function is constructed using the Structural Similarity (*SSIM*) and Peak Signal-to-Noise Ratio (*PSNR*) indices. The GSA-Net network and loss function are utilized to restore images obtained via low-light remote sensing. This proposed method was tested on the Northwestern Polytechnical University Very-High-Resolution 10 (NWPU VHR-10) dataset, and its overall superiority was demonstrated in comparison with other state-of-the-art algorithms using various objective assessment indicators, such as *PSNR*, *SSIM*, and Learned Perceptual Image Patch Similarity (LPIPS). Furthermore, in high-level visual tasks such as object detection, this novel method provides better remote sensing images with distinct details and higher contrast than the competing methods.

**Keywords:** remote sensing image enhancement; global spatial attention mechanism; feature extraction; feature fusion; model compression

## 1. Introduction

The military, earth sciences, agriculture, and astronomy industries are experiencing a surge in demand for high-quality remote sensing images. Nonetheless, less-than-ideal environmental circumstances reduce the brightness and sequester the critical elements in remote sensing images. Since brightness is a major quality component in remote sensing photos, low-light enhancement techniques are required for improved information representation and visual perception [1].

The two image-enhancement techniques are image spatial domain and transform domain methods. Traditional histogram equalization [2] is the most popular image spatial domain algorithm due to its simplicity and efficiency. However, the primary disadvantage of histogram equalization is that if the histogram contains peaks, the generated results are enhanced, resulting in saturation issues and a highly sharpened image. To overcome this issue, histogram-based methods in the spatial domain of the image, such as dynamic histogram equalization [3] and a histogram modification framework [4], have been proposed. Although both can prevent over-enhancement, the details are not emphasized because these methods preserve the input histogram. Recently, adaptive gamma correction with weighted distribution (AGCWD) [5] has been proposed as a new contrast enhancement technique, which produces similar results and may also cause a loss of detail in light areas and an increase in saturation. A previous study proposed a two-dimensional (2D) histogram that uses contextual information to improve the contrast of the input image [6–8]. However, generating a two-dimensional histogram has a high computational cost and is not suitable for many practical applications.

The transform domain method disassembles input images into distinct sub-bands and improves contrast by modifying specific components [6]. In the study by Demirel et al. [9], a singular value equalization method was proposed for adjusting image brightness. The combination of this method with the discrete wavelet transform (Gonzalez et al.) [1] improved the contrast enhancement. Another method, enhancing contrast in remote sensing images with discrete wavelet transforms and adaptive intensity transformations, was proposed by Lee et al. [10]; however, this method requires specific parameter settings, rendering it impractical in the real world. A sub-band decomposition multiscale Retinex method, coupled with a hybrid intensity transfer function, was introduced (Jang et al.) [11] to improve the optical remote sensing images. Also, a generic method of illumination normalization for multiple remote sensing images was proposed (Zhang et al.) [12]. This method first enhanced the contrast in the gradient domain and then adjusted the brightness by equalizing singular values. However, contrast and details are not emphasized because this algorithm focuses primarily on maintaining illumination consistency.

Lore et al. [13] proposed Low-Light Net (LLNet), a deep autoencoder for contrast enhancement and denoising of low-light images caused by accelerated development of deep learning. The authors of [14] introduced transformative neural networks and argued that the conventional multiscale Retinex algorithm is a feedforward transformative neural network with various Gaussian transformation kernels. Retinex-Net is a deep learning method based on Retinex image decomposition [15], and the entire model is implemented using transformative neural networks. It was also used to establish the Low-Light Paired (LOL) dataset under natural conditions. Some cutting-edge end-to-end methods (Han et al., Zhang et al.) [16,17] used U-Net (Ronneberger et al.) [18] as their fundamental structure and added dense residual blocks to each layer to incorporate multiscale information. Although these methods provide competitive performances on benchmark datasets, their lengthy inference time is not conducive to widespread application.

Although deep learning-based algorithms for image augmentation provide encouraging results, some drawbacks, such as blurred details, poor color devotion, and poor visual quality, cannot be ignored. Therefore, based on the current U-Net architecture, the present structured image enhancement model, termed GSA-Net, reduces the loss of spatial information according to sampling. This algorithm builds a U-shaped network using multiscale sampling and introduces global spatial attention (GSA) with respect to image flow, thus enabling the interaction of feature vectors of each branch across channels, suppressing redundant information, and using the feature fusion module to improve the perception of low-scale texture details and multilevel features. Taken together, this method proposes selective kernel feature fusion (SKFF) to effectively integrate the features throughout the reconstruction phase rather than connection approaches to fuse the maps of various resolutions. In summary, the primary contributions of the study are as follows:

- Depthwise separable convolution is a lightweight convolution operation that significantly reduces the number of parameters and computations. Herein, we propose replacing the ordinary convolution in GSA-Net with depthwise separable convolution, reducing the number of parameters from 29.86 M to 7.06 M (a reduction of about 76%).
- A global attention module is introduced to weaken the noise response and integrate local information. Specifically, the global attention mechanism replaces the convolution layers of U-Net and is embedded into the network backbone.
- We propose an improved loss function that combines the peak signal-to-noise ratio (*PSNR*) and structural similarity (*SSIM*) quotient to avoid the model optimization direction deviation and gradient diffusion. This loss function guides the network to train and improve the convergence of the model.
- The proposed model is evaluated based on a synthesized low-light image enhancement dataset, and the results demonstrate that it achieves state-of-the-art performance in image enhancement. Moreover, we facilitate object detection on the enhanced images, which has positive implications for remote sensing images.

## 2. Related Studies

### 2.1. Data Augmentation

With the widespread application of deep learning in computer vision, the diversity of datasets is crucial for the performance of algorithms. In order to enhance the dataset used in our study, we employ Gamma correction as an effective data augmentation technique. The gamma correction possesses unique advantages in adjusting the brightness and contrast of images to augment the dataset for improved training and evaluation of our model.

Gamma transformation is a non-linear process that enhances or suppresses different intensity regions of an image, especially under low-light conditions, to improve the visibility of the details. While determining the parameter values for gamma transformation, we consider the specific requirements for simulating low-light synthetic images. Small gamma values (<1) enhance the details in darker regions, making them distinct, a critical aspect for simulating images in low-light conditions. In addition, we ensure that the gamma value range includes one to preserve the original brightness of the image. Larger gamma values (>1) are applied to suppress the brighter areas of the image, preventing excessive amplification and achieving visual balance. This customized selection of gamma transformation and parameter values caters to the specific demands of simulating low-light synthetic images, enhancing image quality and adaptability in various scenarios. The luminance of the transformed image decreases with the increasing gamma value. Changing the gamma value can affect the quality of remote sensing images. A gamma value of four is used in this study to produce a darker image. The transformed V channel image is then merged back into the V channel of the original image, and the modified image is converted back to RGB space to produce a darkened remote sensing image. The NWPU VHR-10 dataset comprises 650 remote sensing images captured under normal lighting conditions with high contrast and categorized into ten classes. Figure 1 illustrates the NWPU VHR-10 dataset. To enhance the dataset diversity and simulate challenging scenarios, seven sets of parameters are chosen randomly, which generate seven sets of remotely sensed images under weak illumination conditions. Ultimately, this augmentation process yields a comprehensive training set consisting of 4550 images. Consequently, 700 synthetic images are created by combining 100 images with typical lighting that comprise the test set, while the remaining images provide the training data. This method darkens the remote sensing images, providing several low-light images. The transformation formula for gamma is as follows:

$$I_{gm} = \alpha I^{\gamma} \tag{1}$$

when $\gamma > 1$, gamma transformation can be used to darken an image, which facilitates the generation of additional data for a specific dataset. In this study, we present examples where $\alpha = 1$ and the $\gamma$ values are 1.5 and 4. Figure 2 illustrates an example of a synthesized low-light image.

U-Net is a convolutional neural network proposed for biomedical image segmentation tasks [19–22]. The term "U-Net" is derived from the network's structure, which resembles the letter "U". It uses a symmetric encoder–decoder structure and skip connections [23] in the decoder part to merge the feature information extracted from the encoder with that extracted from the decoder, thereby enhancing the reconstructed input image. The encoder component of U-Net comprises convolutional layers, pooling layers, and activation functions, which are used to extract features from the input image. The decoder's reconstruction layers, skip connections, transformative layers, and activation functions are used to reconstruct the output image.

In the field of image enhancement, U-Net is frequently employed to convert low-quality input images to high-quality output images. Specifically, the input image serves as the network's input, whereas the output image is the reconstructed image produced by the network. The U-Net network autonomously learns to convert low-quality images into high-quality ones using the training dataset. Furthermore, U-Net enhances the remote sensing image characteristics, such as contrast, sharpness, and details. Training the U-Net

network is used to obtain an effective model for enhancing the quality of remote sensing images and their practical applications.

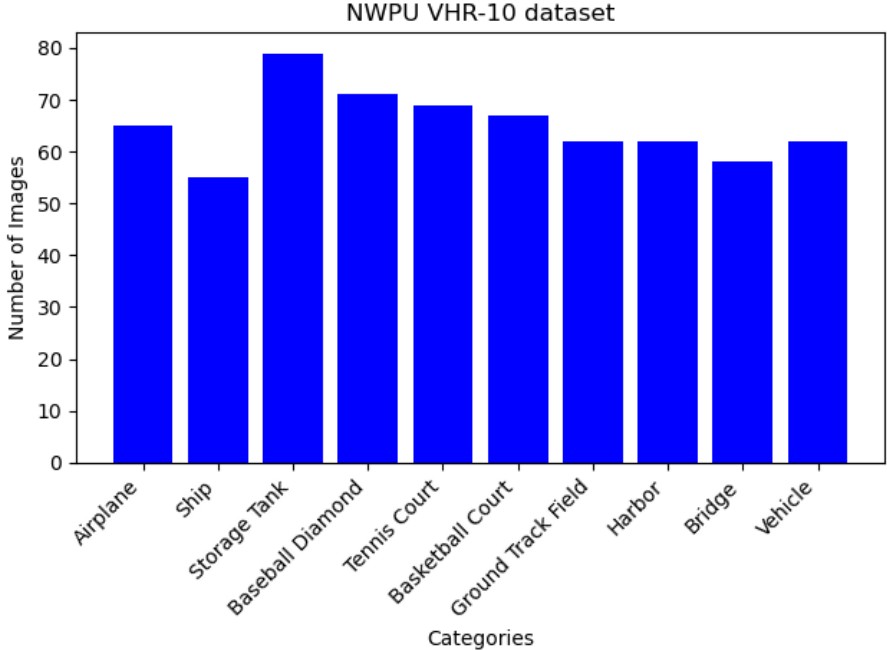

**Figure 1.** NWPU VHR-10 dataset.

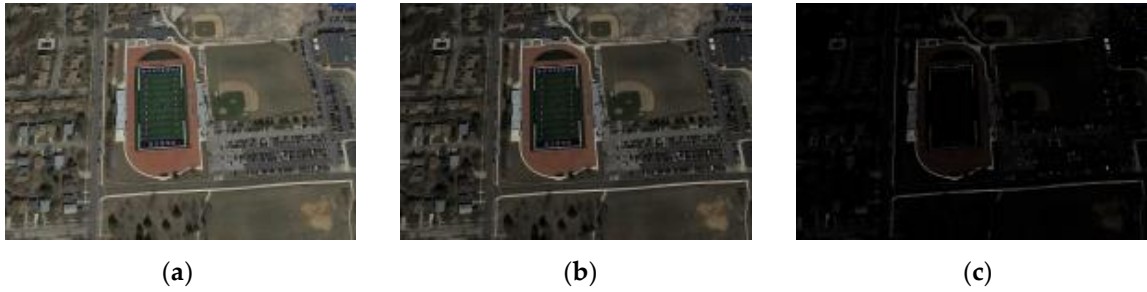

**Figure 2.** Example of synthesizing low-light images. (**a**) normal-light remote sensing images; (**b**) $\gamma = 1.5$; (**c**) $\gamma = 4$.

### 2.2. U-Net

The technological improvements in remote sensing image enhancement are vital, which renders their real-world significance apparent when applied to target detection scenarios. The incorporation of U-Net architecture into the target detection framework yielded a 15% increase in detection accuracy using a dataset of aerial surveillance images. This example illustrates the efficiency of our proposed enhancements in refining the identification and localization of specific targets amidst complex and cluttered scenarios, with potential implications for security and environmental monitoring.

### 3. Proposed Method

Herein, we have introduced the model's primary architecture, followed by an analysis of the function of each module and the loss function's derivation process.

### 3.1. GSA-Net

The multiresolution feature extractor, picture texture reconstruction layer, and feature fusion module constitute the majority of the network. The downsampling [24] and channel attention modules extract spatial details and semantic information and are the components

of the multiresolution feature extractor. Multiple Global Spatial Attention (GSA) modules make up the image texture reconstruction layer, which directs the network to recover the details of the image texture. The feature fusion module [19] functions at the network's end to aggregate the features from various levels in a multidirectional manner and bridge the semantic gap caused by various stages and scales.

The size of the input data for GSA-Net is not rigidly constrained and is typically determined by the characteristics of the task and dataset. It is commonly set as $H \times W$, where $H$ represents the image height and $W$ is the width.

Firstly, for the encoder (downsampling path) of GSA-Net, we employ $3 \times 3$ depthwise separable convolutions to downsample the low-light remote sensing images at the original resolution. Each layer is equipped with GSA to extract comprehensive and rich semantic information. Further details about the GSA block will be elaborated in Section 3.2. Additionally, in the GSA-Net, for the main feature path at the end of the GSA, we utilize the Pixel (Un) Shuffle method as a downsampling module. Subsequently, these feature maps are concatenated with the shallow feature maps obtained from the previous downsampling, and the regular U-Net process is continued. Each downsampling module outputs feature maps with a size of $\frac{H}{2^N} \times \frac{W}{2^N}$, where $N$ is the number of downsampling modules. The middle layers, serving as connecting components between the encoder and the decoder, do not induce significant size changes. In the decoder (upsampling path), each upsampling module increases the size of the output feature maps to $H \times 2^N \times W \times 2^N$ through upsampling and convolution operations, where $N$ is the number of upsampling modules. Finally, the SKFF (Selective Kernel Feature Fusion) method is employed to consolidate information in the decoder (reconstruction process). Figure 3 illustrates the structure of GSA-Net.

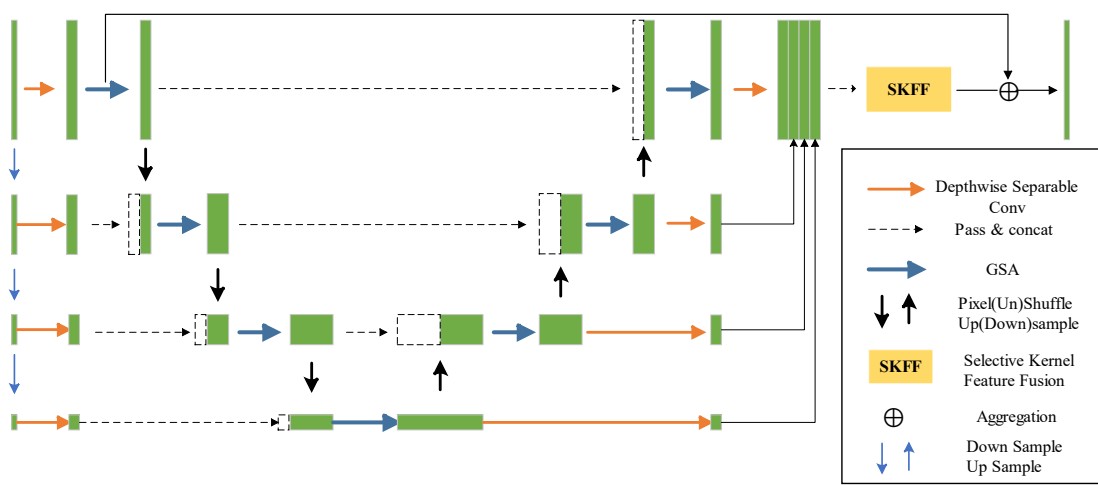

**Figure 3.** Network architecture of GAS-Net.

GSA-Net is distinguished from other methods based on its unique advantages, which are manifested in adopting depthwise separable convolution for a lightweight design and integrating a global attention module. These features reduce the parameter count and enhance local and global image information fusion. However, the network still faces challenges such as computational complexity in large-scale image data, reliance on specific datasets, and sensitivity to hyperparameter choices. To address these limitations, the present study improved the loss function by incorporating an optimization approach using a combination of Peak Signal-to-Noise Ratio (*PSNR*) and Structural Similarity Index (*SSIM*). This targeted optimization enhances the model's convergence, thereby overcoming the constraints and ensuring a robust and reliable performance of GSA-Net in practical applications.

*3.2. GSA Block*

As shown in Figure 4, the top-level subnetwork employs GSA blocks to capture global information, including two depthwise separable convolutions (DSCs) + PReLU layers and

AdaptiveAvgPool2d, AdaptiveMaxPool2d, interpolation (Resize block in Figure 3), and a Spatial Attention (SPA) module. Specifically, based on the input feature map X with size $H \times W \times C$. AdaptiveAvgPool2d and AdaptiveMaxPool2d are used to extract representative information, resulting in an output feature map with dimensions $H1 \times W1 \times C$. Then, the image with global information is upscaled using an interpolation function, followed by Conv + PReLU processing to reduce the channel number, resulting in a global feature map with a size of $H \times W \times C_1$. Subsequently, we apply the SPA block to enhance the attention in different regions in the global feature map. The block also applies both max-pooling and average-pooling in a channel-wise size, and then the two feature maps are subtracted to generate a feature descriptor and highlight the informative regions. Finally, the input feature map (encoding local information) and the optimized global feature map (encoding global information) are combined using the DSC + PReLU function, resulting in an output feature map with a size of $H \times W \times C$.

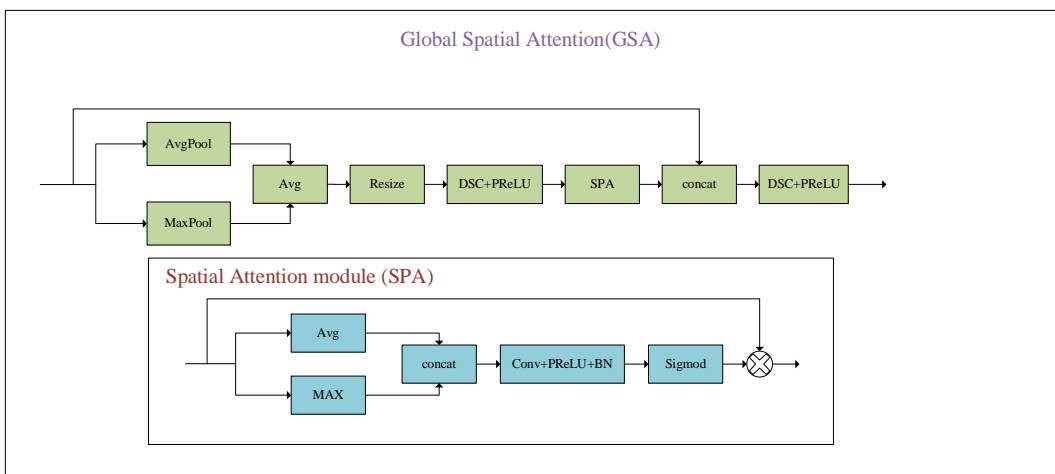

**Figure 4.** Structure of the global attention mechanism.

### 3.3. DSC

DSC is a lightweight convolution operation [25] that splits the traditional operation into depthwise and pointwise convolutions, significantly reducing the number of parameters and computations. The key distinction between depthwise convolution [26] and pointwise depthwise convolution [27] lies in their operational methodologies. Depthwise convolution operates independently on each channel, whereas pointwise depthwise convolution performs linear combinations across channels. Depthwise Separable Convolution (DSC) amalgamates these operations by first employing depthwise convolution, followed by pointwise convolution for inter-channel mixing. Herein, we incorporated DSC into the GSA-Net network for model lightweighting.

In standard convolution, the number of parameters is determined by the size of the convolutional kernel and the number of input channels. In contrast, Depthwise Separable Convolution (DSC) focuses solely on each input channel during depthwise convolution, resulting in smaller convolutional kernel sizes. Pointwise convolution subsequently linearly combines channels through element-wise operations. This design effectively reduces the number of parameters on each channel, and the standard convolution operation with DSC is supplanted, resulting in a 76% reduction in model parameters. This feature enhances the model's computational efficiency but reduces its storage space requirements, rendering it suitable for scenarios with limited resources.

For optimal performance and precision of DSC, selecting the appropriate kernel sizes and numbers for depthwise and pointwise convolutions is essential. In addition, enhanced DSC operations, such as group and deformable pointwise convolutions, can improve the model's accuracy.

Furthermore, DSC reduces the number of parameters and computations while maintaining model accuracy. The depthwise separable convolution employed for model lightweighting is depicted in Figure 5.

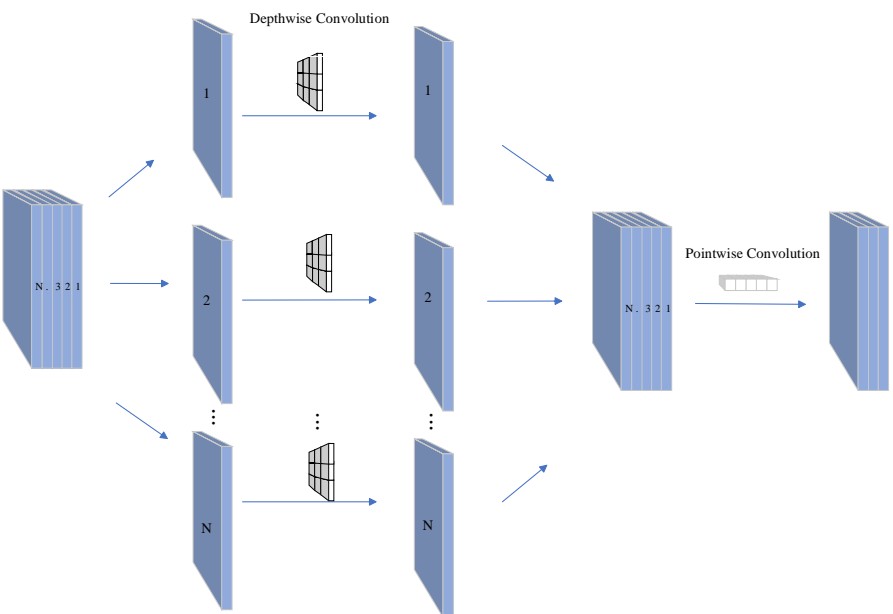

**Figure 5.** The process of depthwise separable convolution for model lightweighting.

### 3.4. SKFF Module

The SKFF module dynamically adjusts the receptive field by two operations, Fuse and Select, as illustrated in Figure 6. Fuse combines the information from multiresolution streams to generate global feature descriptors, while Select uses these descriptors to recalibrate and aggregate the feature map. Specifically, the three branch streams in this study are as follows:

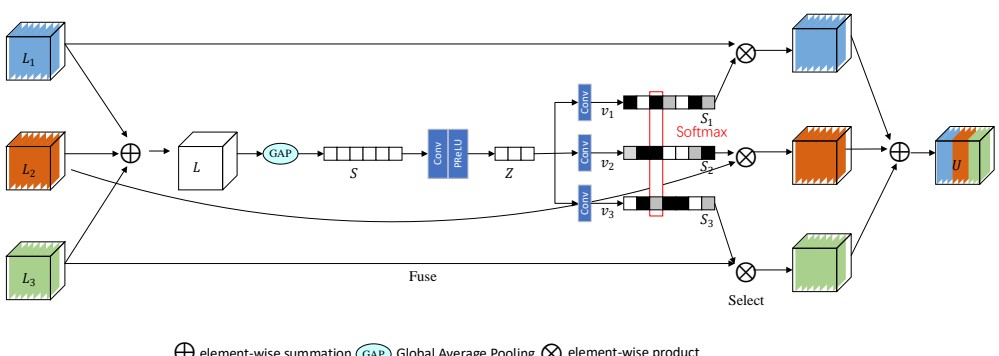

**Figure 6.** Architecture of the SKFF module.

Fuse: The theoretical foundation behind the selection of three parallel convolution streams is rooted in the demand for multiscale information fusion to enhance the model's perceptual capabilities of different scale features. The design aims to introduce convolution streams with distinct receptive fields to capture multiscale contextual information from the input data. Each parallel convolution stream is dedicated to extracting features of specific scales, ensuring the model comprehensively understands the multiscale characteristics of the data. It receives input from three parallel convolution streams and combines the multiscale features through element-wise summation:

$$L = L_1 + L_2 + L_3 \tag{2}$$

Channel $s \in R^{1 \times 1 \times C}$ is computed using global average pooling (GAP) on the $L \in R^{H \times W \times C}$ dimension. Next, a tight feature representation $z \in R^{1 \times 1 \times r}$ is generated using the channel-downscaling Conv layer, where $r = \frac{C}{8}$. Finally, the feature vector $z$ passes through three parallel channel upsampling layers, providing three feature descriptors: $v_1, v_2, v_3 \in R^{1 \times 1 \times C}$.

Select: The SoftMax function is applied on $v_1, v_2,$ and $v_3$ to generate attention activations $S_1, S_2,$ and $S_3$, which are used to recalibrate the multiscale feature maps $L_1, L_2,$ and $L_3$. The process of feature recalibration and aggregation is defined as follows:

$$U = S_1 L_1 + S_2 L_2 + S_3 L_3 \tag{3}$$

The SKFF uses six times fewer parameters than aggregating features through concatenation while still producing better results. The specific structure of the SKFF module is shown in Figure 6.

### 3.5. Loss Function

The goal of training the GSA-Net model is to infer the mapping correlation between low-light image X and normal-light image Y, such that the low-light image can be enhanced to resemble the normal-light image. Currently, the mean squared error (*MSE*) loss function [13] and the mean absolute error (MAE) loss function [28] are the predominant loss functions that measure the error between corresponding pixels in the field of computer vision. The *MSE* is susceptible to outliers, resulting in over-constraint [29], whereas MAE lacks gradient constraints [30], resulting in weak model convergence. Some studies have proposed a structural similarity index (*SSIM*) loss function based on human visual perception [31], which optimizes the model according to visual sensory direction; however, the illuminance and color restoration results are unsatisfactory.

The central concept underlying *SSIM* is the incorporation of subjective human perception. *SSIM* evaluates three variables: (1) distortion is less apparent in very bright regions (luminance); (2) it is less apparent in areas with complex textures (contrast); (3) adjacent pixels form a structure in space that is highly sensitive to the human eye (structure). Consequently, *SSIM* assesses the three variables mentioned above using the following formulas:

$$l(x,y) = \frac{2\mu_x\mu_y + C_1}{\mu_x^2 + \mu_y^2 + C_1} \tag{4}$$

$$c(x,y) = \frac{2\sigma_x\sigma_y + C_2}{\sigma_x^2 + \sigma_y^2 + C_2} \tag{5}$$

$$s(x,y) = \frac{\sigma_{xy} + C_3}{\sigma_x\sigma_y + C_3} \tag{6}$$

In the above formula, $\mu$ represents the mean, $\sigma$ represents the variance, and $C_1, C_2,$ and $C_3$ are constants used to fine-tune *SSIM*. Moreover, $C_1, C_2,$ and $C_3$ satisfy the following correlation:

$$C_1 = (K1L)^2, C_2 = (K2L)^2, C_3 = C_2/2 \tag{7}$$

In Equation (7), $L$ represents the dynamic range of the pixels. For an 8-bit grayscale image, $L = 256$. *K1* and *K2* are <1 and are typically set to 0.01 and 0.03, respectively. Therefore, the formula for calculating *SSIM* is as follows:

$$SSIM(x,y) = [l(x,y)]^{\alpha} \cdot [c(x,y)]^{\beta} \cdot [s(x,y)]^{\gamma} \tag{8}$$

The three factors in Equation (8) have similar effects on subjective perception, and therefore, $\alpha = \beta = \gamma = 1$. The following equation is used to calculate *SSIM*:

$$SSIM(X,Y) = \frac{(2\mu_x\mu_y + C_1)(2\sigma_{xy} + C_2)}{\left(\mu_x^2 + \mu_y^2 + C_1\right)\left(\sigma_x^2 + \sigma_y^2 + C_2\right)} \tag{9}$$

*PSNR* is based on a direct comparison of the differences between pixels. The first step is to calculate the *MSE* of all pixels in two images:

$$MSE = \frac{1}{mn} \sum_{i=0}^{m-1} \sum_{j=0}^{n-1} [I(i,j) - K(i,j)]^2 \tag{10}$$

Taking the logarithm of the result yields the *PSNR*, which is calculated as follows:

$$PSNR = 10 \cdot \log_{10}(\frac{MAX_I^2}{MSE}) \tag{11}$$

In order to constrain the training process, accelerate the convergence speed of the model, and improve the visual quality of the enhanced image, we take into account the characteristics of the aforementioned loss functions. The weighted part of the proposed loss function (Fan et al. 2022) [32] is removed, and that which consists of the quotient of *PSNR* and *SSIM* to predict the error between low-light images and normal images is used, according to the following equation:

$$L_{ps} = \frac{1 - SSIM(X,Y)}{PSNR(X,Y) + \omega} \tag{12}$$

In the equation, *X* and *Y* represent the samples, and $\omega$ is a constant that is usually set to 0.005. This equation avoids the small value of the initial training *PSNR*, which leads to gradient vanishing or explosion while not introducing additional parameters. The combination of *PSNR* and *SSIM* as a loss function exhibits satisfactory general performance and can be extended to similar fields, such as image restoration and denoising.

## 4. Experiments

### 4.1. Experimental Design

To validate the feasibility of the proposed algorithm, we employ the NWPU VHR-10 dataset (Huang et al.) [5] based on the experimental environment shown in Table 1. The Adam optimizer is adopted, with an initial learning rate of 0.0002 and a learning rate decay of 0.00001 after each epoch. The training process is terminated when the learning rate decreases to 0.00001, and the total number of iterations is set to 200. Also, a comparative analysis with state-of-the-art algorithms was conducted in recent years.

**Table 1.** Experimental environment.

| Operating Environment | Detailed Configuration |
| --- | --- |
| System | ubuntu20.04 |
| Processor Model | Intel Xeon Platinum 8255C @ 2.50 GHz |
| Graphics Card | RTX 2080 Ti(11 GB) |
| CUDA Version | 10.1 |
| Deep Learning Framework | Pytorch 1.11.0 |

### 4.2. Dataset

NWPU VHR-10 is a geospatial remote sensing dataset consisting of 650 images with objects and 150 background images, with a total of 800 images, for object detection. The dataset comprises ten object categories: airplanes, ships, oil tanks, baseball fields, tennis courts, basketball courts, athletic fields, harbors, bridges, and cars. Given the small size of the NWPU VHR-10 dataset, the images are converted to the Hue Saturation Value (HSV) color space to mitigate the overfitting problem during the training process and enhance the model's generalization ability. The *V* channel image is gamma-transformed to produce a composite low-light image $V_{ark} : v_{dark} = \alpha V^\gamma$, where $\alpha \in (0.8,1), \gamma \in (1.3,5)$. Then, the *V* channel of the image is replaced with $v_{dark}$, while the other two channels remain unchanged. The image is subsequently converted back into the Red–Green–Blue (RGB)

color space to generate the composite low-light image. For each normal-light image, seven sets of parameters are randomly selected to create seven low-light images, resulting in a total of 4550 training images. Of these, 700 composite images including 100 normal-light images comprise the test set, and the remainder constitute the training set. Table 2 presents seven randomly chosen examples of parameters.

**Table 2.** Seven randomly selected parameter examples.

| Parameter Combination | $\alpha$ | $\gamma$ |
|:---:|:---:|:---:|
| 1 | 0.85 | 2.0 |
| 2 | 0.90 | 3.5 |
| 3 | 0.80 | 1.8 |
| 4 | 0.88 | 4.0 |
| 5 | 0.82 | 2.8 |
| 6 | 0.95 | 1.3 |
| 7 | 0.86 | 3.2 |

*4.3. Evaluation Metrics*

The present study employs six evaluation metrics as criteria to quantitatively evaluate the performance of the proposed low-light image enhancement algorithm. The paper incorporates various metrics to evaluate image generation or processing tasks. These metrics include *PSNR, SSIM, SNR*, normalized mutual information (NMI) (Studholme et al. 1999) [32], learned perceptual image patch similarity (LPIPS) (Zhang et al. 2018) [33], and normalized root mean square error (NRMSE) (Hyndman et al. 2006) [34]. This ensemble of metrics forms a comprehensive set of performance measures. Typically, the numerical range for *PSNR* and *SNR* is typically between zero and positive infinity, with higher values indicating better image quality. *SSIM* values typically range from −1 to 1, with values closer to 1 indicating high image quality. NMI values range from 0 to 1, with higher values indicating better image similarity. LPIPS and NRMSE values range from 0 to positive infinity, with lower values indicating better image quality. *PSNR* reflects the level of image distortion, *SSIM* measures the similarity between two images, *SNR* indicates the *SNR* in the image, NMI reflects the correlation between images, NRMSE measures the error between images based on pixel values, and LPIPS measures the perceptual similarity between images. *PSNR, SNR, SSIM*, and LPIPS are evaluated according to the following formulas:

$$PSNR = 10lg\frac{256^2}{MSE} \tag{13}$$

$$SNR = 10lg\frac{\sum_{i=1}^{M}\sum_{j=1}^{N}x(\beth,j)^2}{\sum_{i=1}^{M}\sum_{j=1}^{N}[x(i,j)-y(i,j)]^2} \tag{14}$$

$$MSE(x,y) = \frac{1}{MN}\sum_{i=1}^{M}\sum_{j=1}^{N}|x(i,j)-y(i,j)|^2 \tag{15}$$

wherein *MSE* stands for mean squared error, while *M* and *N* represent the width and height of the image, respectively. Moreover, *i* and *j* denote the horizontal and vertical coordinates of a pixel; *x* and *y* refer to the sample and the label, respectively.

$$SSIM(X,Y) = \frac{(2\mu_x\mu_y + C_1)(2\sigma_{xy} + C_2)}{(\mu_x^2 + \mu_y^2 + C_1)(\sigma_x^2 + \sigma_y^2 + C_2)} \tag{16}$$

*x* and *y* represent the sample and label, respectively. Moreover, $\mu_x$ and $\mu_y$ denote the means of *x* and *y* and $\sigma_x$ and $\sigma_y$ denote their variances. $\sigma_{xy}$ represents the covariance be-

tween $x$ and $y$, while $C_1$ and $C_2$ are constants that prevent the denominator from becoming too small and resulting in unstable outcomes.

$$d(x, x_0) = \sum_l \frac{1}{H_l W_l} \sum_{h,w} \| w_l \odot (\hat{y}^l_{hw} - \hat{y}^l_{0hw}) \|^2_2 \tag{17}$$

In Equation (17), $d(x, x_0)$ represents the distance between image patch $x$ and $x_0$, $W_l$ is the image feature vector, and $l$ is the layer number. $H$ and $W$ denote the height and width of the image, respectively. $\hat{y}^l$ and $\hat{y}^l_0$, respectively, represent the normalized values of the feature stack and channel unit of the $l^{\text{th}}$ layer.

### 4.4. Qualitative Analysis of Experimental Results

To assess the effectiveness of our low-light image enhancement algorithm, we conduct a comparative study with classical and efficient traditional techniques. These techniques include LIME (Guo et al. 2016) [35] and CLAHE (Yadav et al. 2014) [36]. Additionally, we compare our approach with representative deep learning algorithms, such as SCI (Ma et al. 2022) [37], RRDNet (Zhu et al. 2020) [38], LLFlow (Wang et al. 2022) [39], MIRNet (Zamir et al. 2020) [38], and Zero-DCE (Guo et al. 2020) [40]. Zero-DCE takes the image as input and generates a high-order curve, while SCI achieves self-calibrated illumination learning through weight sharing. Our proposed GSA-Net is also included in the comparison. This method simplifies the design of network structures and enhances images with basic operations. RRDNet is a three-branch convolutional neural network that decomposes input images into illumination, reflection, and noise components, estimates noise accurately, and restores illumination by iteratively predicting loss for denoising. Each algorithm is evaluated in the same experimental setting.

All the algorithms above can address the issue of insufficient illumination in low-light images, and most of the enhancement algorithms can restore object contours and colors effectively. However, the CLAHE algorithm may cause color distortion in the enhanced image, whereas the LIME algorithm may result in uneven color restoration and chaotic hues. Although the RRDNet and Zero-DCE algorithms produce superior visual effects overall compared to the previous two, they are insufficient at reducing noise and artifacts. The output image of the SCI algorithm has insufficient color constraints, resulting in an effect similar to that of a foggy image. The LLFlow and MIRNet algorithms show limited effectiveness in enhancing low-light remote sensing images. Conversely, our proposed algorithm generates augmented images with uniform illumination and contrast, suppressing artifacts and noise and achieving superior subjective visual results compared to other algorithms. Moreover, the comparison of enlarged details in the lower-left corner of the figure reveals higher and more realistic color restoration and precision of the proposed algorithm (Figure 7).

### 4.5. Quantitative Analysis of Experimental Results

To further validate the advanced performance of the proposed model, Table 3 provides quantitative standards for the algorithm described above; the optimal values are highlighted. The GSA-Net's aggregate performance indicators are *PSNR*, *SSIM*, *SNR*, NMI, LPIPS, and NRMSE at 30.110, 0.863, 24.361, 0.833, 0.172, and 0.232, respectively. The results demonstrate that the proposed model has significant advantages over conventional algorithms LIME and CLAHE. The *SNR* and NMI are enhanced by 23.9% and 14.0%, respectively, compared to the traditional deep learning algorithm Zero-DCE. Compared to the LLFlow and MIRNet algorithms, the *PSNR* improves by 28% and 23.7%, respectively. In addition, the proposed model outperforms the state-of-the-art algorithm RRDNet in terms of *SNR* and NMI by 30.6% and 17.0%, respectively. The SCI algorithm's NMI index is similar to that of GSA-Net, while the remaining indicators are inferior to those of the proposed model. In addition, both LPIPS and NRMSE of the proposed model provide the best results, significantly outperforming competing algorithms, indicating that GSA-Net is

capable of learning features that conform to visual patterns. In conclusion, the proposed model achieves low-light remote sensing image enhancement from multiple perspectives and levels with an outstanding performance.

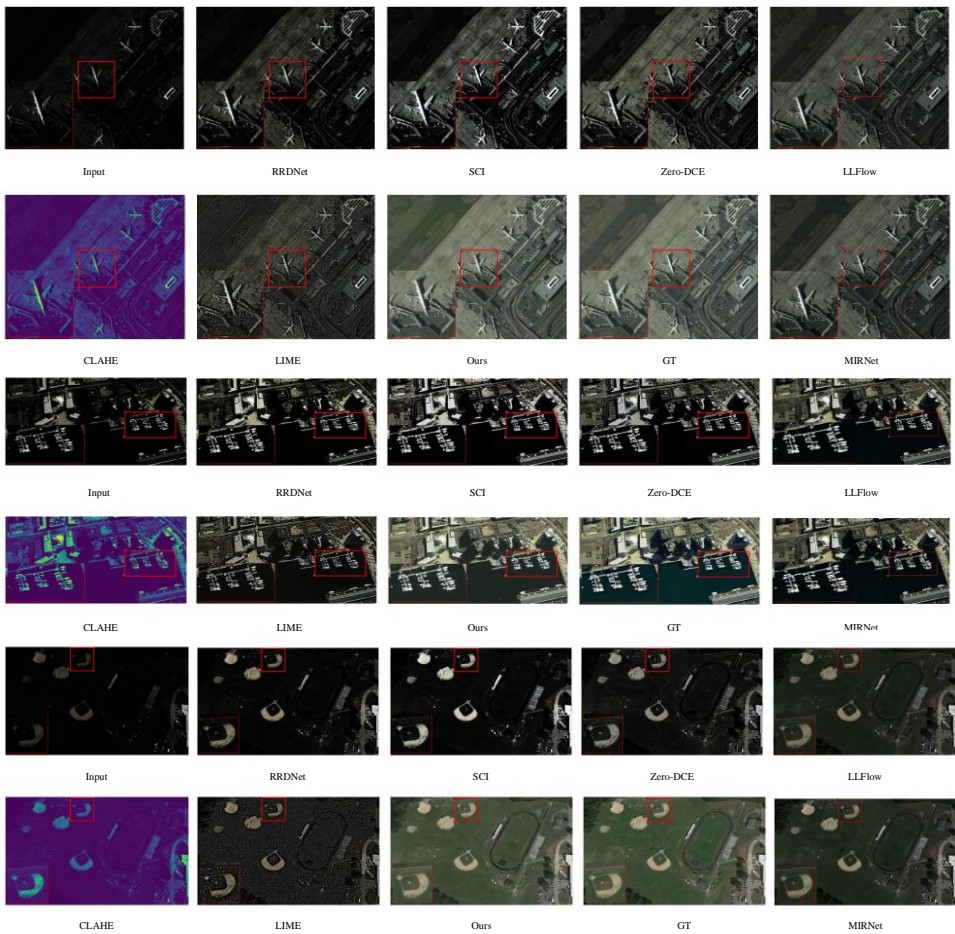

**Figure 7.** Qualitative experimental results.

### 4.6. Loss Experiment

In order to further establish the superiority of the proposed model, the *MSE*, MAE, *SSIM*, the improved MAE loss function (Charbonnier [41]), and the proposed improved loss function based on the GSA-Net model were evaluated using the NWPU VHR-10 dataset. The outcomes are presented in Table 3; the most significant outcomes are highlighted in bold. The *MSE* loss function aligns with the implicit information of the restored image owing to its simplicity, intuitiveness, and its effectiveness in preserving smoothness, coupled with its differentiability. However, its expression includes an exponent that emphasizes the outliers, resulting in poor network convergence. The MAE loss function depicts outliers and performs slightly better than *MSE*, but the discontinuous derivative of the analytical formula hinders the model's convergence ability. Charbonnier enhances the MAE by incorporating a constant to mitigate the gradient leap problem, substantially increasing the *SNR* index. The *SSIM* loss function simulates the updated gradient of the visual system, preserving the image texture details. While some indicators are markedly enhanced, the lack of sensitivity to the mean deviation of bright regions in an image results in undersaturated colors. The proposed loss function strikes an equilibrium between robustness and representability. Despite the fact that *SNR* and NRMSE indicators are slightly inferior, the other indicators outperform those of the comparative loss functions, confirming the improved prediction ability and robustness of the model to achieve the objective of estimating model bias (Table 4).

**Table 3.** The comparison results of different algorithms are presented.

| Evaluation Metrics | Algorithm | | | | | | | |
|---|---|---|---|---|---|---|---|---|
| | **RRDNet** | **SCI** | **ZeroDCE** | **LLFlow** | **MIRNet** | **CLAHE** | **LIME** | **OURS** |
| *PSNR* | 19.378 | 21.356 | 20.784 | 23.516 | 24.339 | 15.632 | 17.413 | 30.110 |
| *SSIM* | 0.542 | 0.526 | 0.612 | 0.786 | 0.795 | 0.356 | 0.456 | 0.863 |
| LPIPS | 0.387 | 0.362 | 0.354 | 0.322 | 0.284 | 0.639 | 0.543 | 0.172 |
| *SNR* | 18.653 | 18.292 | 19.661 | 20.121 | 20.864 | 14.334 | 15.314 | 24.361 |
| NMI | 0.712 | 0.815 | 0.732 | 0.696 | 0.735 | 0.432 | 0.654 | 0.833 |
| NRMSE | 0.276 | 0.258 | 0.268 | 0.263 | 0.254 | 0.563 | 0.388 | 0.232 |

**Table 4.** Evaluation results of different loss functions on the NWPU VHR-10 dataset.

| Evaluation Metrics | Function | | | | |
|---|---|---|---|---|---|
| | *MSE* | **MAE** | **Charbonnier** | *SSIM* | *PSNR/SSIM* |
| *PSNR* | 27.376 | 26.195 | 29.552 | 26.224 | 30.110 |
| *SSIM* | 0.811 | 0.834 | 0.867 | 0.851 | 0.863 |
| LPIPS | 0.245 | 0.342 | 0.189 | 0.263 | 0.172 |
| *SNR* | 21.369 | 20.475 | 24.726 | 23.698 | 24.361 |
| NMI | 0.791 | 0.735 | 0.812 | 0.789 | 0.833 |
| NRMSE | 0.312 | 0.368 | 0.225 | 0.267 | 0.232 |

### 4.7. Ablation Experiment

This study analyzed the effect of GSA structure, SKFF, and DSC on remote sensing image enhancement performance by conducting super-resolution experiments on the NWPU VHR-10 dataset. Table 5 shows the *PSNR* and *SSIM* values, and the parameters of various model variants are compared according to the ablation experiment comparison results. When DSC is eliminated, the network's performance improves marginally, but the number of network parameters increases by 76%. The standard convolution is replaced with a DSC block to obtain a lighter model and improve deployment. The other two enhancement measures can increase the network's image enhancement performance, and their combined use yields the best results.

### 4.8. Case Study

To validate the applicability of our proposed algorithm, we investigate the issue concerning remote sensing object detection under low-light conditions. As shown in Figure 8, YOLOX (Ge et al. 2021) [42] is used to detect objects in low light and restore remote sensing images. The first row of the figure illustrates the detection of airplanes; three airplanes are not detected in the low-light remote sensing image, and a home is mistakenly identified as a ship compared to the restored image. The second row demonstrates the detection of ships and ports, and the house is incorrectly identified as ships and ports in the image on the left, whereas the augmented images are detected accurately. The third row demonstrates the detection of baseball and athletics fields; among these, the athletics field and three baseball fields are not detected in the low-light image on the left, and a house is incorrectly identified as a ship, whereas all detections in the restored image are accurate. These results illustrate the encouraging application potential of our proposed algorithm for object detection via remote sensing.

**Table 5.** Comparison of results from ablation experiments.

| DSC | SKFF | GSA | *PSNR* | *SSIM* | Parameters |
|:---:|:---:|:---:|:---:|:---:|:---:|
| × | × | × | 20.287 | 0.542 | 30.37 M |
| √ | × | × | 18.689 | 0.533 | 6.88 M |
| × | √ | × | 25.645 | 0.758 | 30.07 M |
| × | × | √ | 23.332 | 0.637 | 30.16 M |
| × | √ | √ | 30.230 | 0.876 | 29.86 M |
| √ | √ | √ | 30.110 | 0.863 | 7.06 M |

"×" indicates that this module has not been added, while "√" indicates that this module has been added.

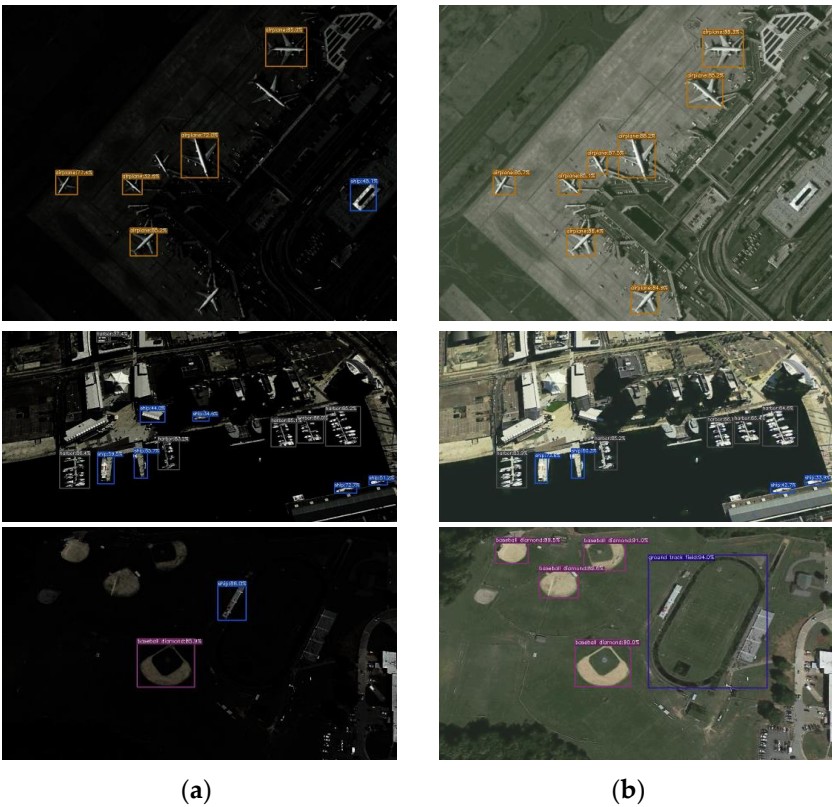

(**a**)　　　　　　　　　　　(**b**)

**Figure 8.** Remote sensing image object detection test experiment. (**a**) Detection of targets in low-light remote sensing images; (**b**) object detection on restored remote sensing images in this study.

## 5. Conclusions

In this study, we proposed a multilevel feature fusion algorithm for improving remote sensing images captured in low-light conditions. Specifically, we employed conv+PReLU layers to generate varied inputs with diverse spatial resolutions and designed the GSA module to capture global data exhaustively. In addition, SKFF was embedded in the model to fuse all information effectively. To aid the network in learning the mapping correlation between purposeful images, we developed a combined loss function to improve the model's color recognition ability and enrich the color of the enhanced images. The experimental results of the analysis on the NWPU VHR-10 dataset demonstrated a superior subjective and global performance of our algorithm compared to the majority of advanced algorithms. The relatively high structural similarity index indicates the applicability of our remote sensing methodology. Thus, future research will concentrate on investigating lightweight models that decrease network space complexity while enhancing the visual effect of enhanced images. We will also strive to improve the existing models to handle various restoration tasks, including image denoising and deblurring.

**Author Contributions:** Investigation and methodology, M.Z. and M.H.; formal analysis, R.Y.; validation and writing—original draft preparation, M.H. and B.L.; writing—review and editing, M.Z. and B.L.; supervision, M.Z. All authors have read and agreed to the published version of the manuscript.

**Funding:** This research was funded by the Innovation Fund of Marine Defense Technology Innovation Center of China: 2022 Innovation Center Innovation Fund Project, under Grant number JJ-2022-712-02.

**Institutional Review Board Statement:** Not applicable.

**Informed Consent Statement:** Not applicable.

**Data Availability Statement:** The data used in this study are publicly available.

**Conflicts of Interest:** The authors declare no conflicts of interest.

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
