# Peer review of "Deep Learning-Based Technique for Remote Sensing Image Enhancement Using Multiscale Feature Fusion"

_sensors, doi:10.3390/s24020673_

Round 1

Reviewer 1 Report

Comments and Suggestions for Authors

Here are my comments:

1. In 3.1 GSA Net, the size of the input data and the size of the feature maps output by each module should be specified, and the downward black arrow should indicate the method used for downsampling and upsampling.

2. In 3.2 GSA Block, the size of the feature maps output by each part  should be specified, such as the average pooled size, which is difficult for others to code.

3. What is the theory why the depth is shortened from S to Z and then increased to S  in the 3.4 SKFF Module? Why do the U have two L2 colors and two L1 colors?

4. I noticed in Table 4 that the accuracy used by all three modules is actually lower than using only SKFF and GSA modules. So what is the significance of using DSC in your network?

5. In the SOTA comparative experiment, the use of methods from the past three years is rare

6. Why is SSIM 0.863 in Table 4 and 0.873 in Table 3? This reduces the credibility of the experiment.

Author Response

Dear Editors and Reviewers:

Thank you for your letter and for the reviewers’ comments concerning our manuscript entitled “Deep Learning-based Technique for Remote Sensing Image Enhancement using Multiscale Feature Fusion” (ID: sensors-2800594). Those comments are all valuable and very helpful for revising and improving our paper, as well as the important guiding significance to our research. We have studied comments carefully and have made correction which we hope to meet with approval. Revised portions are marked in yellow in the paper. The main corrections in the paper and the responds to the reviewer’s comments are as attached files.

Reviewer 2 Report

Comments and Suggestions for Authors

The article introduces a novel multilevel feature fusion algorithm aimed at enhancing low-light remote sensing images. The authors tested this method on the NWPU VHR-10 dataset and demonstrated superior performance compared to existing state-of-the-art algorithms. Here are my comments on specific aspects of the article:

  1. An explanation is needed for the Network Architecture of GAS-Net, as depicted in Figure 3.
  2. It's important for the authors to highlight the Resize block featured in Figure 3.
  3. To enhance clarity, the authors should provide an explanation or a reference distinguishing between depthwise and pointwise convolutions.
  4. The methodology behind calculating the 76% reduction in parameters using DSC instead of standard convolution needs explanation.
  5. The rationale behind choosing three parallel convolution streams in the Fuse operation of the SKFF Module requires explanation.
  6. Referencing is necessary to support the claim that MSE is susceptible to outliers, leading to over-constraint, while MAE lacks gradient constraints, resulting in weak model convergence.
  7. The authors should include a list of the seven sets of parameters that were randomly selected to generate seven low-light images.
  8. Detailed information on the formation of deep learning training and test sets is lacking in the article.

Author Response

(The authors gave the same response as above.)

Reviewer 3 Report

Comments and Suggestions for Authors

Dear Authors,

I have shared feedback within the commented PDF file, highlighting various aspects to consider while preparing the revised version.

Good luck.

Comments on the Quality of English Language

Moderate editing of the English language is required.

Author Response

(The authors gave the same response as above.)

Reviewer 4 Report

Comments and Suggestions for Authors

1. A brief summary

The article proposes a deep learning model for improving remote sensing images in low light conditions. The manuscript is carefully compiled and well-debugged. It is not entirely clear whether the topic fits well for the Sensors journal, but this issue will probably be resolved by the editors. Some minor comments made below, do not reduce the overall positive impression from the work.

2. General concept comments

The research setup uses unnaturally degraded remote sensing data (or maybe not even “degraded”, but only “modified” for the computer vision point of view). At the same time, the authors do not explicitly discuss the limitations of the study related to this fact. Should we expect that the identified advantages of the proposed method will remain so when working with real low-light data? I think this is the critical question.

3. Specific comments

3.1. Line 287: L=255 (maybe 256? it is clear that 8 bits vary from 0 to 255, nevertheless we have 256 levels). The same applies to formula 13.

3.2. In section 4.2: the abbreviation is given first and then its decoding (for example, HSV (Hue Saturation Value)), but in other sections of the article the style is different – first the term is given and then its abbreviation (for example, in the annotation, peak signal-to-noise ratio (PSNR)). It looks illogical.

3.3. Line 374: СLAE - CLAHE ?

3.4. Line 391: is there a mistake when using a comma as a separator for the integer and fractional parts ?

Best regards!

Author Response

(The authors gave the same response as above.)

Round 2

Reviewer 1 Report

Comments and Suggestions for Authors

1. Besides text descriptions, in all network architecture diagrams, all feature maps should have the size  in the form of H/n, such as H/2, H/4.

2. What is the theory why the depth is shortened from S to Z and then increased to S in the 3.4 SKFF Module? Why does the S vector have a process of becoming shorter and then longer, and what is the theory for this.

Why do the U have two L2 colors and two L1 colors? Is it a drawing error or is there any theory?

3. Due to the significant difference in metrics between your method and other compared methods,try to compare one more STOA in past 3 years, some papers provide code that can be found on GitHub.

Author Response

(The authors gave the same response as above.)
